# Mental Health Awareness and Promotion during the First 1000 Days of Life: An Expert Consensus

**DOI:** 10.3390/healthcare12010044

**Published:** 2023-12-24

**Authors:** Hasto Wardoyo, Nila Djuwita Moeloek, Ray Wagiu Basrowi, Maria Ekowati, Kristin Samah, Widura Imam Mustopo, Emi Nurdjasmi, Indah Suci Widyahening, Bernie Endyarni Medise, Febriansyah Darus, Tonny Sundjaya, Bunga Pelangi

**Affiliations:** 1National Family Planning Coordinating Agency (BKKBN), Jakarta 13650, Indonesia; sek.kepala@yahoo.co.id; 2Indonesian Ophthalmologist Association (PERDAMI), Jakarta 10320, Indonesia; nilafmoeloek@gmail.com; 3Department of Community Medicine, Faculty of Medicine, Universitas Indonesia, Jakarta 10320, Indonesia; indah_widyahening@ui.ac.id; 4Health Collaborative Center (HCC), Jakarta 10320, Indonesia; pelangibunga.bp@gmail.com; 5Danone Specialized Nutrition, Jakarta 12940, Indonesia; tonny.sundjaya@danone.com; 6Indonesian Women Empowerment Group (Wanita Indonesia Keren), Jakarta 12980, Indonesia; maria.stefastrigo@gmail.com (M.E.); kristin.samah@gmail.com (K.S.); 7Indonesian Association of Psychologist Special Capital Region of Jakarta (HIMPSI Jaya), Jakarta 12410, Indonesia; widurai@gmail.com; 8Indonesian Midwive Association (IBI), Jakarta 100560, Indonesia; emitaufik@yahoo.com; 9Child Health Department, Faculty of Medicine, Universitas Indonesia, Jakarta 10430, Indonesia; bernie.medise@yahoo.com; 10Obstetric Gynaecology Department, Indonesian President Hospital RSPAD Gatot Subroto, Jakarta 10410, Indonesia; febriansyah.darus@gmail.com

**Keywords:** one-thousand days of life, perinatal mental health, women, children, primary healthcare facilities, Indonesia

## Abstract

The first 1000 days of life constitute a critical phase that will determine the optimum growth and development of a child. An important factor in this phase of life is the perinatal mental health of mothers and children. Mental health awareness is an important public health issue with significant impacts on mothers, spouses, and families, as well as the long-term emotional and cognitive development of children as well. However, the awareness and promotion of mental health within the realms of reproductive health, maternal health, and infant health, i.e., the first 1000 days of life, do not receive high prioritization in Indonesia. Nonetheless, Indonesia, with its existing primary healthcare system, has the potential to raise awareness of and promote the importance of perinatal mental health for its citizens. This experts’ consensus proposes several strategies to maximize the usefulness of primary healthcare facilities in Indonesia, including Community Health Centers and Integrated Healthcare Posts, to support perinatal mental health awareness and promotion during the first 1000 days of life. The success of this program, in return, will improve the health status of women and children in Indonesia.

## 1. Introduction

The growth and development of a child are significantly influenced in the first 1000 days of life, as this period represents the most critical phase, serving as the golden period in which a child can grow and develop optimally [1]. This period comprises 270 days of pregnancy and 730 days within the first two years of life when a child’s brain grows and develops substantially [2,3]. It is also a crucial phase for the development of the whole body, including the metabolic and immune systems [3]. The first 1000 days of life are highly sensitive to environmental factors, including nutrition, as they contribute to proper growth and development or, conversely, can cause numerous long-term health problems, including impaired growth and non-communicable diseases in later life [3,4]. A significant emphasis is therefore put on the first 1000 days of life by ensuring proper nutritional intake for pregnant mothers to optimize fetal formation, growth, and development. Subsequently, the growth of infants up to the age of two years must be continuously monitored as impactful growth disturbance tends to occur during this period [1]. Any disorder that occurs during this period, especially those affecting nutritional intake, will have lasting and difficult-to-correct impacts on the survival and long-term development of children beyond the age of 2 years [1].

These impacts are reflected in the infant mortality rate (IMR), reflecting life expectancy, well-being, and the overall quality of a nation’s health. The United Nations Children’s Fund (UNICEF) revealed that the IMR of Indonesia in 2020 was 16.85 per 1000 live births [5]. The most significant factor contributing to infant mortality is the low birth weight [6,7]. Due to a high prevalence of anemia and chronic energy deficiency among pregnant mothers, low birth weight is commonly observed among Indonesian infants (i.e., 6.2%). Furthermore, inadequate breastfeeding and improper feeding practices and caregiving during the first 2 years of life can disrupt a child’s growth and development [8]. Taken together, the health status of a newborn is closely associated with maternal health [9].

The maternal mortality rate (MMR) serves as another indicator to assess the overall health status of a nation due to its sensitivity to improvements in healthcare services, both in terms of accessibility and quality. Maternal death within this indicator is defined as all deaths during pregnancy, childbirth, and the postpartum period caused by pregnancy, childbirth, and postpartum conditions or their management. This indicator does not include deaths due to other causes, such as accidents or incidents. In 2015, the MMR of Indonesia was recorded at 305 per 100,000 live births (from 390 per 100,000 live births in 1991). This led to Indonesia being declared unsuccessful in achieving the Millennium Development Goal (MDG) of reducing the MMR to 102 per 100,000 live births [7]. In 2017, Indonesia retained a high MMR (i.e., 177 deaths per 100,000 live births), positioning it as the nation with the third highest MMR in Southeast Asia [10]. Furthermore, the 2020 Long Form Population Survey reported that the MMR of Indonesia was at 189 deaths per 100,000 live births during pregnancy, childbirth, or the postpartum period [5].

The perinatal period, spanning from the start of pregnancy to the first year postpartum, is a transitional phase that can induce anxiety and stress in women [11]. The onset of maternal depression is triggered by hormonal changes and can impact the mental health of mothers, disrupting their basic and health needs [12]. According to the World Health Organization (WHO) in 2022, one in five women globally will experience poor mental health conditions during pregnancy and postpartum [13]. The WHO also stated that globally in 2019, 10% of pregnant women and 13% of women who had recently given birth would experience mental disorders, especially depression. In developing countries, this prevalence was even higher, reaching 15.6% during pregnancy and 19.8% after delivery [14]. This poor mental health could be attributed to poor health or various health challenges faced by women, infants, or even their families [13].

Poor maternal mental health has adverse effects on the physical and psychological health of both women and infants [15]. Furthermore, low perinatal mental health in women can result in reduced attendance for Antenatal and Postnatal Care, inadequate nutritional coverage, increased risk of pre-eclampsia, elevated risk of breastfeeding difficulties, and poor parenting practices [16]. Based on the 2018 Basic Health Research (“Riset Kesehatan Dasar/Riskesdas”) in Indonesia, mothers who did not attend Antenatal Care with healthcare providers were 2.4 times more likely to experience depression compared to those who received proper antenatal care [17]. Thus, antenatal depression is a significant burden on public health and one of the most common mental disorders during the perinatal period [18].

Perinatal depression is a widespread public health issue that disproportionately affects women with lower incomes and negatively impacts parenting outcomes and child development [19]. Maternal mental health issues, such as prenatal anxiety, can induce vasoconstriction and increase blood pressure and heart rate. This results in reduced blood supply to the placenta and increased health risk to the fetus [20]. Prenatal anxiety affects pregnant women and leads to clinical symptoms, including agitation, cognitive distortion, constant worry, shortness of breath, palpitations, and restlessness [21]. Mood disorders during the antenatal period are also associated with an increased risk of pre-eclampsia, linked to heightened overall causes of death [22]. Depressed mothers also have a higher risk of giving birth prematurely and are 2.8 times more likely to give birth to neonates with low birth weight compared with mothers with good mental health [23,24].

Depressed mothers can also exhibit lower parenting capabilities [25]. Caring for a newborn is a demanding task in which parents will experience a range of emotions [26]. During the infant’s nurturing phase, parental behavior significantly influences the child’s health, including mental health status [27]. Nurturing and caring for children with affection is akin to laying a strong foundation for them to develop the social and emotional skills necessary for a healthy, happy, and prosperous life [28]. This involves providing warm and gentle interactions, helping children feel protected, comforting them, and addressing their basic needs [29]. In contrast, stressed and mentally ill caregivers/parents might have reduced abilities to provide consistent and responsive parenting for their children [30]. The implications of poor mental health conditions during the perinatal period include increased risks of mortality, reduced exclusive breastfeeding rates, diminished bonding and closeness between mothers and infants, elevated risks of stunting and low birth weight in children, elevated risk of poor cognitive development, increased occurrence of diarrhea and other illnesses, increased rate of low immunization, increased rate of suicide, and elevated risk of mental health issues during adolescence [16].

Of note, approximately 7.2 million cases of stunting in developing countries are attributed to psychosocial factors, including perinatal mental health disorders. The main risk factor is maternal depression, accounting for 3.2 million stunting cases [31]. A study in Indonesia reported that maternal depression during pregnancy is significantly associated with stunting occurrences, with 24 stunted children (33.8%) birthed by mothers with a history of depression [32]. Stunting has garnered increased attention due to its long-term effects, leading to children being unable to achieve their full growth potential, having lower cognitive abilities, and being more susceptible to diseases [33]. As the prevalence of stunting in Indonesia is concerningly high, i.e., 21.6% based on the results of the 2022 Indonesia Nutritional Status Survey [34], the perinatal mental health status should be prioritized to help reduce the prevalence of stunting.

Thus, the perinatal mental health status during the first 1000 days of life becomes an essential component that requires attention as it can be interpreted as a contributing factor to the overall health status of mothers and children [35]. It is indeed a critical issue for public health with significant impacts on mothers, spouses, families, as well as the long-term emotional and cognitive development of infants [36]. However, the mental health awareness and promotion within the realms of reproductive health, maternal health, newborn health, child health, and adolescent health have not received high prioritization in low- and middle-income countries, including Indonesia [16].

In Indonesia, primary healthcare services are delivered through 10,321 community health centers (“Pusat Kesehatan Masyarakat/Puskesmas”) across 7230 sub-districts. However, the services are not yet optimized to effectively cover the population residing in 76,941 villages and 8506 urban neighborhoods. Hence, strengthening the networks and linkages of community health centers is imperative through capacity building and an efficient referral system [37]. Integrated healthcare posts (“Pos Pelayanan Terpadu/Posyandu”), as an extension of community health centers, play a crucial role in improving health status, including reducing MMR and IMR, promoting healthy living behaviors, enhancing community participation and capacity in health-related activities, and encouraging family planning [38]. Each integrated healthcare post is operated by at least five community healthcare workers, officially recognized through a formal decree issued by a local governmental unit. The roles and responsibilities of these community healthcare workers encompass various tasks, including data collection, education, mobilization, companionship, and community empowerment. The active engagement of community healthcare workers and continuous support from the local government unit is important to run the integrated healthcare posts routinely. The target groups of the integrated healthcare posts are (i) pregnant women, (ii) postpartum and breastfeeding mothers, (iii) infants, toddlers, and preschool children, (iv) school-age children and adolescents, and (v) the productive age group and the elderly. However, the integrated healthcare posts in Indonesia have not performed mental health assessments for these groups as they focus more on physical examinations. In addition, the competency to conduct mental health screening is insufficient at this primary level [39].

Taken together, there is an opportunity to perform perinatal mental health assessment, intervention, and improvement as primary healthcare services in Indonesia. An experts’ discussion was subsequently conducted to contemplate how to maximize primary healthcare services, particularly at the integrated healthcare post, to promote perinatal mental health awareness for mothers and infants. The results were proposed as an expert consensus to raise mental health awareness and promotion of primary healthcare services during the first 1000 days of life in Indonesia.

## 2. Materials and Methods

The consensus was constructed based on a narrative review followed by a meeting among 21 experts. The narrative review, based on published studies, provided an overview of the condition of mental health in Indonesia related to maternal and child health and opportunities to improve the condition. This review was subsequently disseminated to all experts prior to the meeting.

### 2.1. Narrative Review

A comprehensive search of English- and Bahasa Indonesian-based publications was conducted. The following search terms were used: “perinatal mental health”, “stunting”, “Integrated Healthcare Post”, “1000 days of life”, “impact of mental health to pregnancy”, “social cultural of pregnancy in Indonesia”, “social support”, “mental health and public health”, “mental health service”, “screening of mental health”, “self-rating questionnaire 20/SRQ20”, “policy of mental health”, and “policy of maternity services.” The chosen populations were pregnant women, lactating mothers, postpartum women, and women with infants under 2 years old. The materials included the impact of metal health disorders on maternal and child health, the urgency of mental health services in the first 1000 days of life, and the roles of integrated healthcare post in Indonesia.

### 2.2. Expert Selection

Twenty-one experts from various fields were invited, i.e., six from psychology, one from obstetrics/gynecology, one from pediatrics, one from a mental health organization, one from community medicine, one from mass media, one from health policy, one from occupational health, and eight from the government (i.e., representatives of Ministry of Health, People’s Representative Council, Ministry of Home Affairs, Ministry of Women Empowerment and Child Protection, the National Population and Family Planning Board, Mayor of Semarang, Stunting Reduction Implementation team of West Java, and the Women Empowerment and Child Protection office in North Sulawesi). The eligibility criterion for the experts was that they had at least ten years of work experience in their respective fields.

### 2.3. Procedures and Data Analysis

Relevant studies regarding mental health awareness or intervention in the first 1000 days, particularly in Indonesia or Southeast Asia countries, were scarce. In addition, differences between social norms and cultures between countries render it unlikely to simply transfer and apply related scientific results from other countries to Indonesia, as mental health issues are significantly influenced by social and cultural factors. Therefore, a consensus method (via a modified nominal group/expert panel technique) was used to provide a collective opinion with an expectation that the consensus will serve as a base to conduct more studies in Indonesia on mental health awareness in the first 1000 days [40,41]. The onsite meeting included a presentation and discussion among experts regarding maternal mental health and how to integrate mental health assessment at the integral healthcare post and other primary healthcare facilities. Of note, some experts also provided their remarks via Google Forms. The expert meeting was recorded and analyzed by the authors.

## 3. Results

The expert meeting revealed potential mental health issues during the first 1000 days of life: eating disorders, attention-deficit/hyperactivity disorder, emotional disturbance, trauma, anxiety, depression, sadness, emotional instability, sleep disorder, fatigue, postpartum depression, and developmental disorder among infants. It is not surprising that mental health issues can cause emotional instability that negatively affects the acceptance of pregnancy until delivery, adaptation to the role of a new mother, parenting process, relationship with spouse, family relationship, cognitive functionality, and concern for self-well-being. In particular, maternal mental health issues can incite a feeling of guilt, self-harm intention, or even suicidal ideation. Of note, approximately 60% of breastfeeding mothers in Indonesia reported unhappiness with the breastfeeding process during the pandemic. Nonetheless, 44% of the breastfeeding mothers felt at ease with their breastfeeding experience due to the support from their families and spouses. Indeed, approximately 90% of breastfeeding mothers stated that they required support from their husbands, and 59% also required support from other family members during the breastfeeding process [42].

The subsequent discussion resulted an expert consensus to improve nine aspects of mental health awareness and promotion within the first 1000 days of life: (i) concept of mental health integration; (ii) procedure of mental health assessment; (iii) instrument of mental health assessment; (iv) resources; (v) referral mechanism; (vi) supporting programs; (vii) supporting policy; (viii) thematic campaign; and (ix) collaboration (summarized in Figure 1).

### 3.1. Concept of Mental Health Integration

Keys to improving perinatal mental health services include mental health awareness and recognition that women during the perinatal period are susceptible to mental health issues. Both should be applied within the community as well as by healthcare providers. It is also acknowledged that health intervention is more effective if the preventive and promotive aspects are emphasized. These approaches will facilitate effective budgeting and programming as well. These interventions comprise the identification of high-risk behaviors, identification of supporting behaviors, screening procedures, follow-up, and educational strategies. Regarding mental health awareness, there are at least three important steps to be considered: (i) screening mental health, (ii) educating all parents about mental health concepts and awareness, and (iii) managing and referring patients with mental health issues.

It is paramount to emphasize that there is no health without mental health, as mental health is a basic need. Integration of mental health services during the first 1000 days of life, arguably, would be an effective program for women and children. Maternal mental readiness to be pregnant, deliver, breastfeed, and care for her child (together with her spouse) is crucial to women’s and children’s health. The Regulation of the Ministry of Health number 21/2021 on the Implementation of Health Services for the Period of Pre-Pregnancy, Pregnancy, Childbirth, and Postpartum, as well as Contraceptive and Sexual Health Services, stated that mental health issues or disorders experienced by pregnant mothers would not only affect themselves but also influence the growth and development of the fetus during the pregnancy, postpartum, infanthood, childhood, and adolescence.

In Indonesia, various cultural practices, including pregnancy rituals, exist in many ethnic groups. These practices reflect the local wisdom of social support for pregnant mothers. Pregnancy rituals are interpreted as a form of social support provided within the context of cross-cultural therapy. They involve a series of prayers and family support during each ceremonial process, including emotional support, recognition, information sharing, instrumental support, and social network support [43]. Social support is a way of demonstrating affection, concern, and appreciation for others. Vice versa, individuals who receive social support will feel appreciated and worthy of being part of their social units [44]. During the postpartum period, which lasts approximately two months, the mothers and neonates commonly stay with their families, who assist with household chores, maternal self-care, and childcare [45]. Mothers who receive strong family social support are less likely to experience postpartum depression because they tend to feel at ease and comfortable, leading to a sense of calmness [46]. In addition, maternal social support during pregnancy can also improve the Apgar score among newborns, ensure normal birth weight, reduce postpartum depression, and improve maternal and infant health [47].

High social support reduces the likelihood of experiencing antenatal depression. Pregnant women who receive low social support are more likely to experience mental health issues compared to those who receive high social support. Pregnant women with low social support may lack someone to talk to, crucial information and advice, or individuals who could help alleviate negative emotions associated with challenging situations [48]. There is a significant relationship between low social support and antenatal depression, antenatal anxiety during pregnancy, and self-harm. Therefore, it is important to provide social support for pregnant women, particularly from their immediate family members and the surrounding community. This support can help pregnant women to strengthen their stress coping skills [49].

Perinatal mental health promotion should be developed with a social approach and a community-based support system. The program should provide comprehensive mental health promotion, including (a) respectful service to pregnant women and new mothers, (b) psychoeducation, (c) assessment of mental health, (d) referral system, (e) adequate treatment based on the proper diagnosis, (f) creation and enhancement of the supporting group, (g) full involvement of parents or caretaker in providing care for the maternal–child relationship, (h) improvement of life skills; and (i) stress management. These programs should also target vulnerable populations, such as female youth, females with mental health problems, and marginalized groups.

### 3.2. Procedure of Mental Health Assessment

Mental health assessment is necessary before conception as data on maternal mental health is highly important during the first 1000 days of life. The chosen instrument must be accurate and depict existing healthcare resources and access for mothers. The mental health assessment should be conducted by trained personnel and can be digitalized as a further improvement to replace paper records.

The screening can be done at the integrated healthcare post by empowering the community healthcare workers. This is according to the policy framework outlined in the Regulation of the Ministry of Health number 21/2021, stating that mental health assessments for pregnant women can be conducted during visits to primary healthcare facilities [11]. Mental health screening can be integrated into prenatal and infant health checkups through clinical interviews by inquiring about risk factors of mental health disorders and the individual’s history of mental health issues.

In assessing the maternal mental health status, it is important to consider the individual’s lifelong health history and mental well-being, as well as how these factors influence the mother’s mental state during the perinatal period [50]. Symptoms of depression or anxiety in the pre-conception period are effective predictors for the pre- and postpartum phases [51]. One crucial period that requires attention is adolescence. According to the 2022 Indonesia—National Adolescent Mental Health Survey on adolescents aged 10–17 years, poor mental health is a prevalent issue among Indonesian adolescents, with one in three adolescents (~15.5 million individuals) experiencing mental health problems within a 12-month period and in twenty adolescents (~2.5 million individuals) fulfilling the criteria for a mental disorder [52]. Adolescent girls have higher vulnerability with a greater prevalence of depression (6.7%) compared to adolescent boys (4%). Older adolescents (14–17 years old) intriguingly have a higher prevalence of anxiety disorders (27.2%) compared with younger adolescents (10–13 years old; 26.3%). This aligns with the prevalence of depression, which is 7.7% among older adolescents and only 3.2% among younger adolescents. The highest functional impairment of adolescent mental health is from the domain of family relationships (83.9%). This suggests that adolescents tend to have relationship issues with primary caregivers and difficulties spending time with family. Of note, less than 3% of adolescents with mental health issues have accessed mental health services within a 12-month period, suggesting that most issues go unnoticed and can have dire consequences in later years [52].

Mental health assessments for pregnant women should be conducted, at least in the first and third trimesters. If any issue or disorder arises, the patients should be re-evaluated at the subsequent visit. Routine screening and general psychosocial support from maternal and child healthcare providers are essential during the perinatal period. This ensures that women feel capable of discussing and managing their mental health issues. If mental health conditions cannot be addressed adequately, specialized services tailored to their condition will be required, including promotion of mental health and prevention of poor mental health during the perinatal period, identification of mental health issues, and provision of treatment and referral if required [13]. Taken together, integrating mental health services with antenatal care and screening for pregnant women’s potential depression are crucial for promoting perinatal maternal mental health and reducing potential harm to the mother and fetus.

### 3.3. Instrument of Mental Health Assessment

The Regulation of the Ministry of Health number 21/2021 states that one method for detecting mental health issues is the Strength and Difficulties Questionnaire 25 (SDQ-25). If the anamnesis suggests symptoms of mental disorders, further assessment of reproductive-age couples above the age of 18 years can be performed with the Self Reporting Questionnaire 20 (SRQ-20). The SRQ-20 can be supplanted with illustrations to improve the interviewer’s and interviewee’s understanding of the questions. The mental health assessment can be conducted through an interview at various healthcare facilities, including integrated healthcare posts, community health centers, midwifery clinics, psychology clinics, health clinics, and hospitals. Of note, it will be useful to develop a self-screening method to assess one’s mental health. Hence, there will be more individuals who utilize the instrument to assess one’s mental health status. The assessment can be regularly performed once per month to integrate it with monthly healthcare services at the integrated healthcare post.

### 3.4. Resources

The lack of identification and treatment of women with poor mental health can be attributed in part to a shortage of specialized mental health professionals, particularly in low- and middle-income countries [13]. In 2021, the number of psychiatrists in Indonesia was only 1053 individuals, suggesting that one psychiatrist serves approximately 250,000 people. This ratio is far from the WHO standard, which requires an ideal psychiatrist-to-population ratio of 1:30,000 [53]. Second, it can be attributed to insufficient mental health training for healthcare providers. Hence, they possess limited knowledge and skills in recognizing and counseling mothers on their mental health issues [54]. Healthcare providers also tend to prioritize the physical rather than mental aspects of mothers [54,55,56]. Third, it can be attributed to the limited time of healthcare providers, resulting in the relegation of maternal mental health issues [57]. Fourth, prioritizing and funding mental health issues are indeed lower than those for physical health issues, particularly in low- and middle-income countries [36,58]. Of note, in 2022, 45.14% of municipalities across Indonesia had implemented family health interventions, comprising six antenatal care visits, birth at healthcare facilities, infant healthcare services, toddler growth and developmental surveillance, reproductive health services for prospective couples, elderly health services, and mental health screening. Among those seven services, the mental health screening for individuals > 15 years, did not achieve optimal coverage [39].

Thus, capacity building becomes imperative to enable all related personnel, including social workers, healthcare providers, psychologists, or psychiatrists, to detect early symptoms of mental health deterioration and provide follow-up advice/treatment or refer them to higher healthcare facilities [59]. In addition, the personnel must familiarize themselves with the concept of mental health integration during the first 1000 days of life, including the socialization of mental health awareness and the integration of mental health assessment, as well as the know-how to utilize the assessment’s instrumentation and observational techniques. Improving these relevant skills is also important for medical doctors working at primary healthcare facilities to ensure they can accompany the trained personnel to administer mental health interventions.

Mental health assessment can be conducted through two approaches. First, the survey is performed by trained personnel. Second, the mental health assessment can be self-performed. The advantage of having self-testing is that the individual would be aware (or more aware) of their mental health status, allowing them to manage their own emotions upon recognizing any mental health issues. Hence, the screening process should be easy to perform, accessible, and easy to understand. If necessary, the assessment can be performed by the patient’s family member or close friend; however, it still respects the patient’s data privacy. Of note, it will be important to emphasize to societies that mental health issues are common and not disgraceful (i.e., removing the stigma of mental health). Hopefully, through this campaign, results of the self-performed mental health assessment can be immediately forwarded to the healthcare providers and any incurred issue will be managed accordingly.

### 3.5. Referral Mechanism

The trained personnel at the integrated healthcare post should be able to assess maternal mental health during the first 1000 days of life. Healthcare providers must discuss with pregnant women their social support networks and monitor the mental health status of expectant women. The necessary mental health support for women during the perinatal period can be provided during routine consultations within the maternal and child healthcare services [13]. Within these healthcare services, women can receive information on stress management and support from their relatives and family members. Some women may encounter prolonged mental health challenges during this period and will require additional assistance from maternal and child healthcare services, including professional mental health counseling and referrals to higher healthcare facilities, if necessary.

If any mental health issue is identified, the individual can be managed initially by doctors at the primary healthcare facilities that provide mental health services. Subsequently, the individual can be referred to higher healthcare facilities if required. This emphasizes the urgent need for developing and strengthening the network of mental health services from community healthcare posts to healthcare centers to referral hospitals [60]. In addition, information on available mental health services at primary and referral healthcare facilities must be made as widely available as possible to the communities.

### 3.6. Supporting Programs

Mental health promotion will be effective if complemented by various supporting programs. The first program must promote collaborative educational interventions, a continuous examination process, nutritional and psychological counseling, and improved childcare. The second program will promote mental health comprehensively and massively. The third program enhances social support for mothers and infants during the first 1000 days of life. Of note, working mothers and housewives are vulnerable to various mental health issues. Thus, they will need continuous support from their spouses and families. In agreement with this, the fourth program must increase and strengthen the spouse’s roles in parenting. Indeed, children benefit socially and emotionally when fathers are more engaged in their lives [61]. Of note, healthcare providers can facilitate the formation of supportive peer groups for pregnant women to ensure they receive better social support. In line with this, relevant stakeholders and policymakers should consider the development of community-based social support programs for pregnant women that can be effectively integrated into the maternal healthcare services [48]. The fifth program must strengthen parenting with the positive deviance approach. This indicates that parents must be responsive toward their children’s needs and provide love and care for their children. The parents should learn to communicate positively and stimulate and play with their children. This will create a good parent–child attachment and provide security for children, hence they will be willing to explore and develop their cognitive abilities. The sixth program will increase the capacity to set up family planning. This will improve the physical and mental health of bachelors and facilitate proper and responsible actions once married. The seventh program must promote a socio-health-based preventive program. An example is to minimize the patriarchal culture that considers women as reproductive objects only. Other examples include providing education in sexual and reproductive health and prevention of child marriage and unwanted pregnancy.

### 3.7. Supporting Policy

Implementing laws to promote mental health awareness within the first 1000 days of life is also important. It is important to ensure that a law is available to promote mental health awareness within the first 1000 days of life. A new law (i.e., Law number 17/2023) was recently ratified to regulate all healthcare activities in Indonesia, including mental health awareness, with a strong emphasis on disease prevention [62]. In this law, mental health is defined as an integral part of overall health, in which efforts to achieve optimal mental health must be made through promotion, prevention, treatment, and rehabilitation by the government and the community. Subsequently, governmental regulations, presidential regulations, and regulations of the Ministry of Health are being developed by the executive branch of the government to interpret and implement Law number 17/2023, particularly on perinatal mental health issues in Indonesia. This will facilitate the nationwide interpretation and implementation of activities regulating mental health. In addition, budget advocacy should be promoted based on relevant programs and regulations. In sum, this will ensure continuous implementation of mental health-regulating activities with sufficient funding.

### 3.8. Thematic Campaign

A thematic campaign can be useful to gain attraction and support for mental health awareness within the first 1000 days of life. Several themes of interest include (i) the importance of mental health for mothers and infants during the first 1000 days of life, (ii) the reduction and elimination of mental health stigma, (iii) the positive perception of mental health for women, and (iv) improved awareness regarding seeking out mental health intervention during the first 1000 days.

### 3.9. Collaboration

The penta-helix collaboration (i.e., between the academic, business, community, government, and media) should be initiated and encouraged because mental health is an important issue for all parties. In addition, this type of collaboration should be encouraged from the lowest to highest levels of administration. Next, integration and partnership to strengthen familial bonds can be achieved within the existing system, including integrated healthcare posts and academic institutes. Finally, mental health promotion should be integrated with the care for adolescent health services at the community health center. This approach has been promoted by the Indonesian Pediatric Society as it will facilitate the early detection of mental health issues (at the integrated healthcare post level) and immediate referral to the care for adolescent health services when required.

## 4. Conclusions

Perinatal mental health is important during the first 1000 days of life. Women’s and children’s mental health should be prioritized in Indonesia, as these will significantly contribute to the overall health status of women and children. The primary healthcare facilities in Indonesia, including integrated healthcare posts and community health centers, are useful resources to promote perinatal mental health awareness and provide relevant intervention if necessary. The improved status of perinatal mental health, in return, will improve the health status of women and children in Indonesia.

## Figures and Tables

**Figure 1 healthcare-12-00044-f001:**
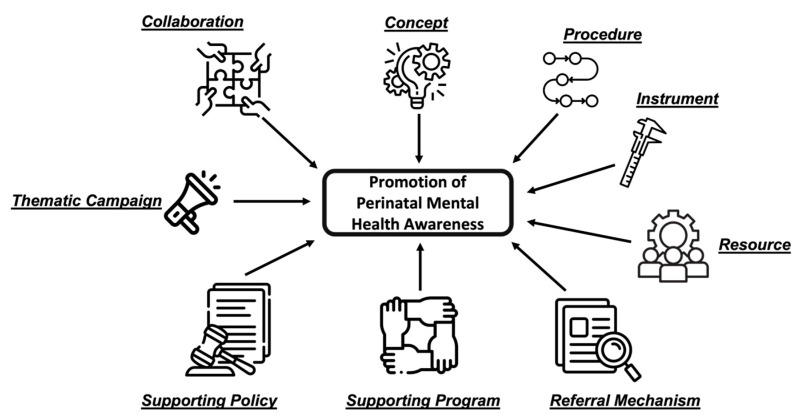
Promotion of perinatal mental health awareness in nine categories. Actions must be performed in the following categories to raise mental health awareness and promote it: (i) concept of mental health integration; (ii) procedure of mental health assessment; (iii) instrument of mental health assessment; (iv) resources; (v) referral mechanism; (vi) supporting programs; (vii) supporting policy; (viii) thematic campaign; (ix) collaboration.

## Data Availability

Not applicable.

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
