# Peer review of "Mental Health Awareness and Promotion during the First 1000 Days of Life: An Expert Consensus"

_healthcare, 2023, doi:10.3390/healthcare12010044_

Round 1
Reviewer 1 Report
Comments and Suggestions for Authors
Thank you for the opportunity to review this manuscript. Overall, it provides a thorough overview of the proposal for improving mental health awareness and promotion in Indonesia.
Some comments as follow:
1. The whole article and proposal is based on the theory of first 1000 days of life affecting subsequent health development. It would be good to include a description and summary of the current evidence of this theory so readers have a clearer idea on the rationale of the proposals in this manuscript. How does it affect the growth and development? Why 1000 days?
2. In Page 5 Line 223 to 227 “Older adolescents had a higher prevalence of anxiety as compared to younger adolescents….. Taken together this suggests a trend that mental health conditions increase with progression of age”. I don’t think comparing the two groups (14-17 vs 10-13 years old) can give a conclusion that mental health condition increase with age progression.
3. Page 10 Line 469 section Supporting policy “It is important to ensure that the law is available to promote mental health awareness”. What sort of law does the author mean? Why the law is important? What is the current situation? The paragraph needs more elaboration.
Thank you.
Author Response
Dear Reviewer
Thank you for your robust reviews and input, please find attached the responded points in rebuttal letter file.
Regards,
Ray W Basrowi

Reviewer 2 Report
Comments and Suggestions for Authors
This manuscript provides a summary from an experts' consensus regarding the improvement of perinatal mental health care during the first 1000 days of life. Evidence that the first 1000 days of life poses particular challenges for maternal mental health is presented in the introduction, along with evidence that maternal mental health has a significant impact on infant development as well as public health.
Due to the breadth of impact the first 1000 days of life has on society, this study brings an awareness and ideas for improvement that could be life-changing. However, the methodology is lacking and needing revision.
Expert consensus is not a commonly used methodology and should therefore be explained more thoroughly in the procedures section. Rationale should be provided. Perhaps a meta-analysis should be conducted instead.
The results section should be reorganized, with the review of the literature being moved to the introduction. The expert consensus ideas should be explored in more depth in the results section.
Author Response

(The authors gave the same response as above.)

Round 2
Reviewer 1 Report
Comments and Suggestions for Authors
The authors had adequately addressed my concerns, and I recommend acceptance on this manuscript. Thank you,
Reviewer 2 Report
Comments and Suggestions for Authors
The revisions look sufficient and help the manuscript's organization be more meaningful to the reader. The only remaining revision I recommend is in the introduction: the heading "3. Results" was mistakenly pasted and should be removed.